# Silent voices: Uncovering women's absence in veterinary surgery publications

**Kelly L. Bowlt Blacklock**[1]*, **Jill R. D. Mackay**[1], **Poppy Bristow**[2], **Ishita Parakh**[3], **Alina Paczesna**[3], **Fiona Mackay**[3], **Kathryn Pratschke**[1]

**1** The Royal (Dick) School of Veterinary Studies, University of Edinburgh, Edinburgh, United Kingdom, **2** Dick White Referrals, Cambridgeshire, United Kingdom, **3** School of Social and Political Science, University of Edinburgh, Edinburgh, United Kingdom

\* kelly.blacklock@ed.ac.uk

## Abstract

### Objective

To determine the factors associated with an author being female in Veterinary Surgery journal publications from 2002 to 2023.

### Study design

Retrospective observational study.

### Sample population

A total of 2881 Veterinary Surgery papers published between 2002 and 2023.

### Methods

Author gender was inferred using the Gender API and verified manually where necessary; only binary categories (male/female) were assigned. Statistical models (frequentist and Bayesian logistic regressions) were used to predict author gender based on publication year, author order (first, second, last), and surgical emphasis.

### Results

Overall, 36% of the authors were female, with 43% of first authors, 37% of second authors, and 28% of last authors being women. The proportion of female first authors increased from 29% in 2010 to 60% in 2022, while female last authors ranged from 10% in 2002 to 36% in 2023. The likelihood of an author being female was influenced by the publication year, being the last author, and the subject of orthopaedics.

**Data availability statement:** All relevant data for this study are publicly available from the OSF repository (https://osf.io/rpb3z).

**Funding:** KBB is indebted to the support of the Royal (Dick) School of Veterinary Studies, who funded this project (Ref: 20028001). The funder did not play any role in the study design, data collection and analysis, decision to publish, or preparation of the manuscript.

**Competing interests:** The authors have declared that no competing interests exist.

## Conclusion

Compared with the demographics of the profession, women remain underrepresented in *Veterinary Surgery* authorships, particularly in senior author positions and orthopaedic papers.

## Disciplinary significance

The underrepresentation of female authors highlights persistent gender disparities in veterinary surgery research. Further research is required to understand the factors that influence female authorship, so that institutions and journals can implement targeted initiatives to promote gender equity.

## Introduction

Scientific progress relies on the diversity of voices contributing to the global body of knowledge. The underrepresentation of women in science has far-reaching implications, affecting career trajectories, scientific advancement, and equitable distribution of opportunities [1,2]. Women face numerous obstacles in academia, including biases in peer review, limited access to resources, and fewer leadership roles [3,4]. Addressing gender disparities is not merely an ethical imperative, but a strategic necessity for advancing research excellence and ensuring the integrity of scientific inquiry [5,6].

In surgical disciplines, including veterinary surgery, these disparities are particularly persistent. In the UK and USA, the veterinary profession was historically male dominated from the early 1960s up to the late 1980s, whereupon women achieved parity in the profession [7]. From the late 1980s onwards, the number of women entering the profession has risen and women now comprise nearly 80% of new graduates and 60% of the workforce in the UK, Canada and USA [8–11]. Despite their numerical dominance, female veterinarians occupy fewer high-ranking academic and policy-related positions compared to their male counterparts [12,13]. A gender pay gap is also prevalent within the profession, with female veterinary surgeons earning lower salaries from graduation onwards [11,14,15].

Publishing scientific papers is crucial for advancing knowledge, establishing professional credibility, and achieving career progression in academia. However, there is considerable gender inequity in the authorship of scientific publications, with women significantly underrepresented in prominent author positions [16–22]. Additionally, publications with female first and last authors tend to attract fewer citations compared to those with male authors in the same roles, further disadvantaging the voices of women within the scientific community [23].

The objective of our study was to examine factors associated with female authorship in *Veterinary Surgery,* the official journal of the American and European Colleges of Veterinary Surgeons (ACVS and ECVS). In the absence of systemic gender effects, we would expect a representation of at least 60% female authors. We hypothesised that female veterinary surgeons would be underrepresented in *Veterinary*

*Surgery* publications, and that gender disparity among authors in *Veterinary Surgery* would not improve over time. We anticipated that this disparity would be particularly notable among last authors [24], and in surgical fields traditionally perceived as 'masculine,' such as orthopedics and large animal surgery [25–27].

## Materials and methods

This project was approved by R(D)SVS Human Ethic Review Committee (Ref: HERC 23_057)

### Publication dataset

Our dataset consisted of *Veterinary Surgery* journal papers from Volume 31, Issue 1 (January 2002) to Volume 52, Issue 4 (May 2023), along with 28 early access papers. Papers were collated using the online database Scopus [28], as well as manual searching of the *Veterinary Surgery* website [29]. Paper titles, abstracts, publication date and author information were extracted. Author information included the names and affiliations of the first, second, and last authors (referred to as 'author order'). The primary author (KBB) divided the papers into 'Small Animal', 'Large Animal', 'Research', 'Exotics', or 'Other' categories (referred to as 'animal category'). Papers within the 'Small Animal' category formed over two-thirds of the published papers, therefore they were further subdivided into 'Orthopedics' or 'Soft Tissue' (referred to as 'surgical emphasis') (KBB). The 'Other' category encompassed papers with surgical relevance that did not fit into our primary classifications, including basic research-focused articles, exotic animal surgery, and multidisciplinary studies. We retained this category to avoid forced classification and to preserve the diversity of topics within the dataset. Although relatively small in number, these papers were included in the analysis to ensure completeness.

Author first names were essential for our name-based author gender categorisation scheme. Therefore, for authors listed with only an initial, we searched for their full first names using their affiliation details and populated these names into the dataset manually. We limited our analysis to first, second, and last authorship positions as these are typically associated with the most prominent academic contributions and career advancement opportunities. Middle author positions vary widely in meaning across disciplines and regions, and are less consistently tied to measures of academic recognition. Future studies may examine full authorship networks to further contextualise these trends.

### Population demographics

*Veterinary Surgery* is the official journal of the American College of Veterinary Surgeons (ACVS), the European College of Veterinary Surgeons (ECVS), and the Veterinary Endoscopy Society (VES). Therefore, to compare our author demographics with that of the ECVS and ACVS, we requested the first names, registration date and surgical emphasis (small animal vs large animal) of diplomates registered with each college.

### Name-based assignment of author gender

We used the publicly available probabilistic database 'Gender API' [30] to assign a binary gender category (male or female) to the first name of each author. Each gender assignment was accompanied with an accuracy rating from 0–100%, based on Gender API's probability rating. If the assigned gender had an accuracy of less than 70% [31], we manually reviewed the author's name and determined their binary gender based on their affiliation details and the pronouns featured in their publicly available professional profiles (e.g., institutional websites, LinkedIn, ResearchGate). Manual gender assignment was conducted independently by two reviewers (KBB and IP), and cross-checked for bias control. We did not attempt to infer non-binary or transgender identities due to limitations of both the API and publicly available information, and we acknowledge this as a limitation of our binary gender framework. If an individual had no discoverable profile or pronoun, and Gender API could not assign a gender, that author was excluded from the modelling. In total, gender could not be determined for 24 authors (0.3% of the dataset).

## Model building and selection

We were interested in determining which factors (publication year, author order, surgical emphasis) were associated with an author being female. We decided to fit statistical models with gender as the response variable. Male was coded as the default and Female as the change of interest.

We analyzed each variable as a fixed effect because they are expected to be constant across individuals [32]. We ran each model as a frequentist binary logistic regression and a Bayesian estimation of probability [33,34]. We considered a range of criteria to establish the optimal model (S1 Table). In our final model, we used a logistic regression to predict author gender with publication year, author order, surgical emphasis, and an interaction between publication year and author order. We described the Bayesian effects using the SEXIT framework, which returns the minimal and optimal required information to describe model's parameters under a Bayesian framework [35].

All analyses were performed in R (version 4.2.3 ("Shortstop Beagle")) and made use of the 'tidyverse' [36], 'broom' [37], 'broom.mixed' [38], 'report', 'easystats' [39] and 'yardstick' [40] packages. Models were fitted via the 'stats' package in R and 'rstanarm' package [41]. We estimated relations via the 'modelbased' package [42].

## Results

A total of 2881 papers were collected from *Veterinary Surgery* from Volume 31, Issue 1 (January 2002) to Volume 52, Issue 4 (May 2023), along with 28 early access papers. Overall, we collected 2881 first author names, with 3% of papers authored by a single author. Thereafter, 2481 papers (86%) included a second author, and 2791 papers (97%) had last author names. Details pertaining to papers assigned to each surgical emphasis are shown in Table 1.

## Veterinary Surgery Authors are predominantly male

First, we wanted to determine the overall landscape of author gender. Across all 2881 papers published in *Veterinary Surgery* between 2002–2023, we found that 36% of authors were female. We determined that female author numbers varied by author order: 43% of first authors, 37% of second authors, and 28% of last authors were female (Table 2). The proportion of female authors fluctuated over the years, ranging from a minimum of 29% in 2010 to a maximum of 60% in 2022. Similarly, the representation of female last authors ranged from a minimum of 10% in 2002 to 36% in 2023. (Fig 1).

To determine whether author gender reflects the demographic of the ECVS and ACVS, we obtained membership information from the ECVS. The ECVS Office provided information pertaining to diplomate gender, registration date and surgical emphasis. For reasons of GDPR, no first names were provided. We received no response for requests for membership data from ACVS, and therefore we obtained limited data about ACVS member demographics from the 2024 ACVS member needs assessment [43]. ECVS membership demographics are reported in detail by Pratschke *et al* [44]. Across the 20 years evaluated for ECVS Diplomate membership (2002–2022), 34% of diplomates were female. Younger ECVS diplomates were more likely to be female, with 50% of new diplomates being female in 2018. Gender distributions among ECVS diplomate specialisms were skewed; only 20% of diplomates specialising in Orthopedics were female, compared to 39% in general specialties and 40% in Soft Tissue specialties. For ACVS diplomates, gender balance is generally

**Table 1. Categories assigned to 2881 Veterinary Surgery papers published between 2002-2023. The category 'other' includes papers pertaining to research or exotic animal species.**

| Category | Type | Number of papers | Percentage (%) |
|---|---|---|---|
| Surgery | Small Animal: Soft tissue surgery | 798 | 27.7 |
| Surgery | Small Animal: Orthopedics | 1120 | 38.88 |
| Surgery | Large Animal | 848 | 29.43 |
| Surgery | Other | 115 | 3.99 |

**Table 2. Distribution of author gender in 2881 Veterinary Surgery papers published between 2002 to 2023, including partial data for 2023. The table presents paper numbers and percentages of male and female authors, as well as published author order.**

| Author Order | Author Gender | Number of papers | Percentage (%) |
|---|---|---|---|
| First | Female | 1242 | 43.11 |
| First | Male | 1633 | 56.68 |
| First | Undetermined | 6 | 0.21 |
| Second | Female | 930 | 37.48 |
| Second | Male | 1537 | 61.95 |
| Second | Undetermined | 14 | 0.56 |
| Last | Female | 791 | 28.34 |
| Last | Male | 1996 | 71.52 |
| Last | Undetermined | 4 | 0.14 |

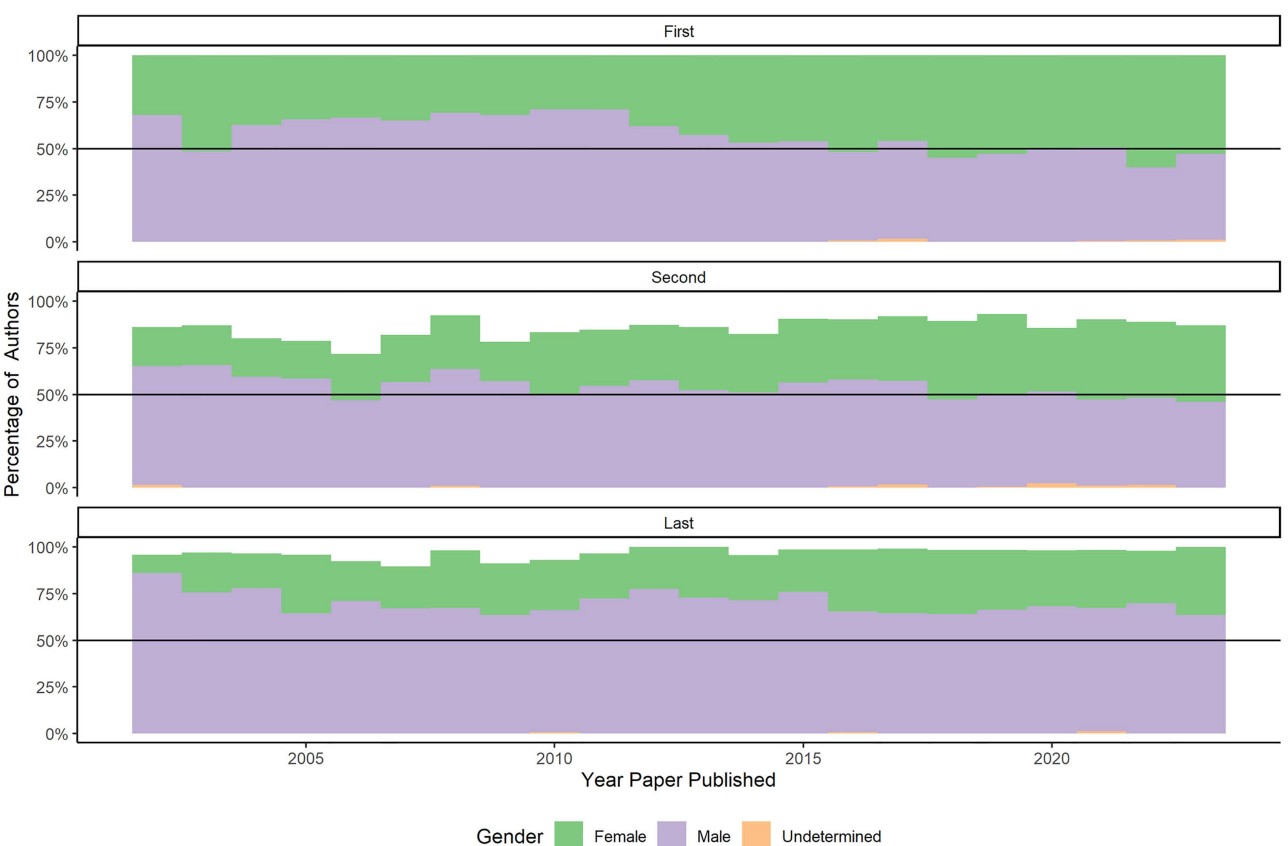

**Fig 1. Proportion of Veterinary Surgery authors assigned a gender of female versus male, shown by year published and author place (first, second, last).** n = 2881 papers. 50% proportion shown.

observed with 48% female and 52% male representation in 2024. Notably, younger ACVS diplomates (under 45 years old) are more likely to be female (51%), whereas older diplomates (45 years and over) are predominantly male (71%).

Therefore, we accept our hypothesis that female veterinary surgeons are underrepresented as authors in *Veterinary Surgery* publications, particularly as last authors. This underrepresentation aligns with broader disparities observed in ECVS (and possibly ACVS) diplomate demographics, particularly where older diplomates are more likely to be male.

## Second and last authors are significantly less likely to be female

Next, we wanted to identify predictors of author gender and achieved this using the explanatory variables: year of paper publication (with 2002 coded as Year 1), author order (first, second or last), and surgical emphasis (small animal: orthopedics, small animal: soft tissue surgery, large animal, other).

The overall explanatory power of our model was low ($R^2 = 0.05$), indicating that the variables included—publication year, author order, and surgical emphasis—account for only a small proportion of the variance in author gender. This low $R^2$ suggests that important factors influencing authorship trends are not captured in the model. These may include systemic influences such as mentorship access, institutional culture, editorial policies, or individual career stage—all of which merit further investigation.

Our model predicted a baseline probability of an author being female at 34%, corresponding to the intercept at the year 2002, being a first author, and publishing in the 'Small Animal: Soft tissue surgery' category.

From 2002 onwards, each year increased the odds of an author being female by 5% (95% CI 4%, 6%). However, the probability of this increase being significant or substantial was 0%, indicating that the year alone is not a reliable predictor of an author's gender [35] (Table 3).

We found that author order closely correlated with author gender (Table 3). Being a second author was associated with a 17% reduction in the probability of that author being female (95% CI −0.44, 0.08), with a 74.5% probability that this is a significant effect, and a 0.42% probability of being a large effect. Being a last author was associated with a 30% reduction in the probability of that author being female (95% CI −45%, −11%) with a 97.7% probability that this is a significant effect, and a 7.1% probability of being a large effect.

In conclusion, we can be confident that second and last authors are significantly less likely to be female than first authors in our dataset, suggesting that more senior authors are more likely to be male.

**Table 3. Model parameters for the binary logistic regression model predicting author gender (Female = 1) using a Bayesian framework with year, author order, surgery category, and an interaction between year and author order. Model estimated using MCMC sampling with 4 chains of 2000 iterations and a warmup of 1000 iterations.**

| Parameter | Median Effect | Odds Ratio | 95% Lower CI | 95% Upper CI | Probability of Direction | Rhat | Effective Sample Size | Prior Distribution | Prior Location | Prior Scale |
|---|---|---|---|---|---|---|---|---|---|---|
| **NA** | −0.67 | 34% (Probability) | −0.86 | −0.49 | 1 | 1 | 1768.29 | normal | 0 | 2.5 |
| **Year** | 0.05 | 1.05 | 0.04 | 0.06 | 1 | 1 | 1760.83 | normal | 0 | 0.42 |
| **Author Place (Second)** | −0.18 | 0.83 | −0.44 | 0.08 | 0.91 | 1 | 1812.58 | normal | 0 | 5.44 |
| **Author Place (Last)** | −0.35 | 0.7 | −0.59 | −0.1 | 1 | 1 | 1672.33 | normal | 0 | 5.27 |
| **Interaction between Year and Second Author Place** | −0.01 | 0.99 | −0.03 | 0.01 | 0.72 | 1 | 1814.75 | normal | 0 | 0.4 |
| **Interaction between Year and Last Author Place** | −0.03 | 0.97 | −0.05 | −0.01 | 1 | 1 | 1671.3 | normal | 0 | 0.38 |
| **Surgery Category: Small Animal Orthopaedics** | −0.53 | 0.59 | −0.64 | −0.41 | 1 | 1 | 2936.44 | normal | 0 | 5.13 |
| **Surgery Category: Large Animal** | −0.01 | 0.99 | −0.14 | 0.11 | 0.58 | 1 | 3018.17 | normal | 0 | 5.49 |
| **Surgery Category: Other** | 0.37 | 1.45 | 0.12 | 0.61 | 1 | 1 | 3959.59 | normal | 0 | 13.1 |
| **Interaction between Year and Second Author Place** | −0.01 | 0.99 | −0.03 | 0.01 | 0.72 | 1 | 1814.75 | normal | 0 | 0.4 |
| **Interaction between Year and Last Author Place** | −0.03 | 0.97 | −0.05 | −0.01 | 1 | 1 | 1671.3 | normal | 0 | 0.38 |

NA: not applicable.

## Underrepresentation of female authors has not significantly improved over time

Next, we were interested in whether gender differences in author order changed over time. Our model showed that the interaction between year and second author position reduced the probability of a given author being female by 1% (95% CI −2%, 1%), with a 0% probability of being either a significant or large effect. We also found that the interaction between year and a last author position reduced the probability of a given author being female by 3% (95% CI −5%, -<1%), but with a 0% probability of being a significant or large effect. Therefore, we would not be confident that these interaction terms differ from the baseline prediction (Table 3).

In conclusion, we accept our hypothesis that gender disparity among authors in *Veterinary Surgery* has not improved over time. This contrasts with trends observed in ECVS and ACVS demographics, where younger diplomates are increasingly likely to be female, and underscores the persistent challenges in achieving gender equity within veterinary surgery research and publications.

## Female authors are significantly underrepresented in Small Animal Orthopedics

Finally, we were interested in whether author gender correlated with surgical emphasis. Publications in the surgical emphasis 'Small Animal: Orthopedics' decreased the probability of a given author being female by 41% (95% CI −47%, −34%), and this effect had a 100% probability of being negative and significant, and 37.0% probability of being a large effect. Publications in the surgical emphasis 'Large Animal' had a 1% decrease in the probability of a given author being female (95% CI −12%, 12%), but this effect had only a 57.3% probability of being negative, 10.7% of being significant, and 0% of being large. Therefore, we cannot be confident that the surgical emphasis 'Large Animal' differs from the baseline of the 'Small Animal: Soft tissue surgery'. Interestingly, publications assigned to the surgical emphasis 'Other' increased the likelihood of a given author being female by 45% (95% CI 13%, 86%), and this effect had a 99.9% probability of being positive, 98.8% of being significant, and an 8.4% probability of being a large effect.

To support model interpretation, we generated conditional averages on the response scale based on the levels of the model. While the proportion of first and second place authors who are female increases over time, this trend is slower for last authors (Fig 2).

In conclusion, female authors are particularly underrepresented in small animal orthopedic publications. The same is not true for 'Large Animal' or 'Other' (research, exotics etc) surgical categories. Therefore, we accept our hypothesis that the underrepresentation of women would be more pronounced in specializations perceived as 'masculine,' such as orthopedics, but reject this hypothesis for large animal surgery. This finding may partially reflect diplomate demographics, as only 20% of ECVS diplomates and 30% of ACVS diplomates specializing in orthopedics are female.

## Summary of key results

In summary, our analysis reveals that women remain underrepresented in *Veterinary Surgery* publications, especially in senior (last) authorship positions and in orthopaedic surgery papers. Although female first authorship has increased over time, gender disparities among last authors persist. The statistical model used explained only a small part of the variation in author gender, which suggests that other unmeasured factors are likely to play an important role. Our findings point to the need for more comprehensive efforts to understand and address the systemic barriers that may influence both opportunities for authorship and the visibility of that work within the scientific literature.

## Discussion

Despite significant strides towards gender parity in the veterinary profession, our findings reveal persistent gender disparities in *Veterinary Surgery* publications, particularly evident in senior author roles and orthopedic specializations. This gender disparity is also present at higher professional (specialist) levels. Over the past two decades, ECVS diplomate membership averaged 34% female, with a notable skew in certain specialties such as Orthopedics. ACVS membership

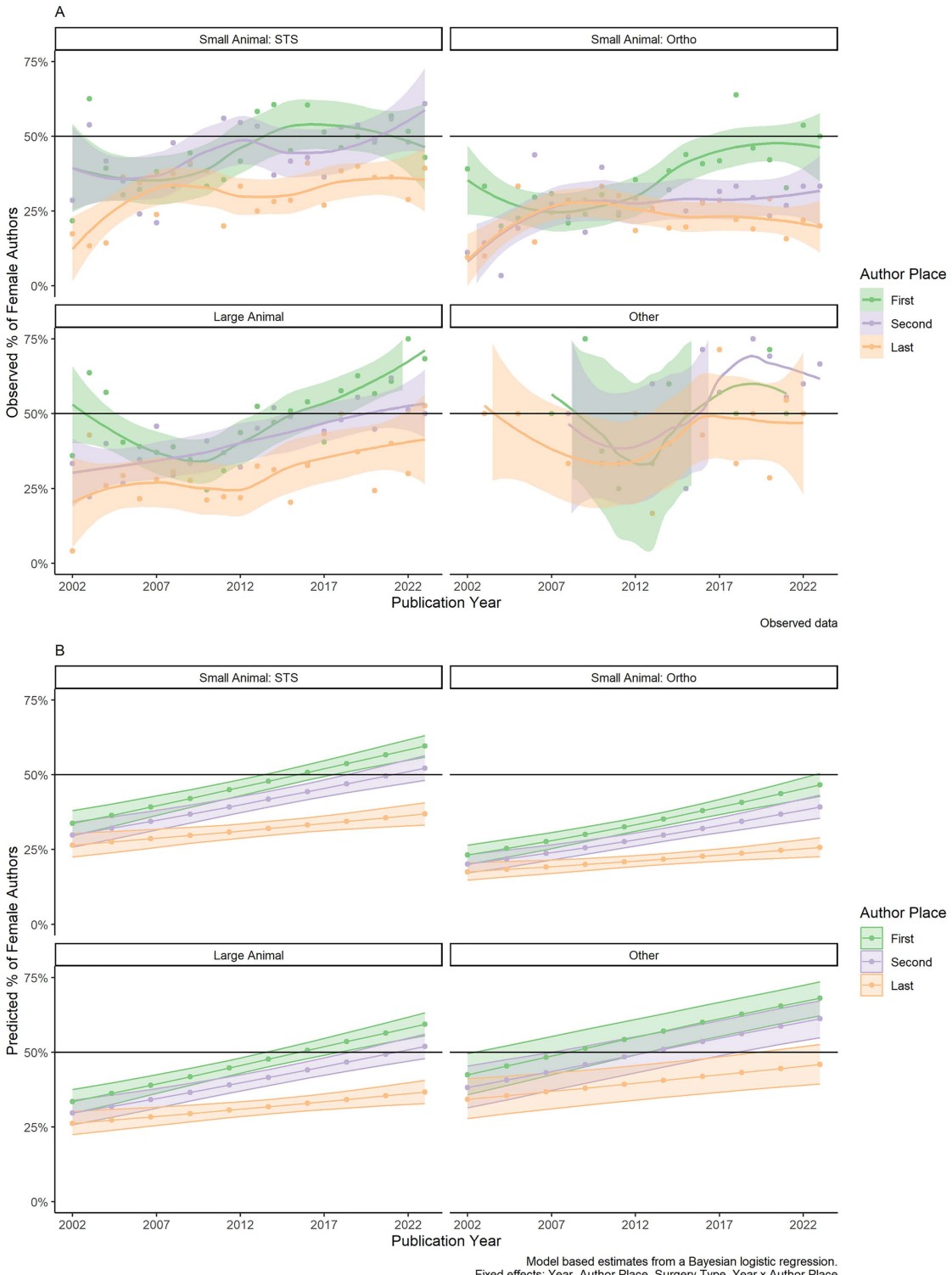

**Fig 2. Observed (A) and model-based (B) trends in female authorship over time.** Panel A displays the raw observed proportions of female authors by year and author position. Panel B presents model-based estimates generated via Bayesian logistic regression, illustrating the predicted probability of an author being female over time, accounting for surgical category, year and author order. These model-based predictions help smooth annual variation and identify broader trends not immediately evident in raw data. Confidence intervals are shown to indicate the level of uncertainty in the estimates.

Model estimated using MCMC sampling with 4 chains of 2000 iterations and a warmup of 1000 using gender as a conditional average and confidence intervals shown (Bayes $R^2 = 0.05$).

data, though limited, shows a balanced gender distribution overall in 2024, but with distinct age-related trends favoring younger female diplomates. The modest explanatory power of our models ($R^2 = 0.05$) suggests that factors such as publication year, author order, and surgical specialty only partially account for the gender disparities observed in authorship. This low $R^2$ implies the presence of unmeasured variables, including structural and cultural forces operating within the academic and surgical landscape. For instance, previous research is suggestive of gendered patterns in academic collaboration: men may be more likely to engage in homophilous, instrumental networks aimed at career advancement, while women are more inclined toward inclusive, mentoring-oriented collaborations [45,46]. However, these differences in style do not emerge in isolation—they are shaped by institutional norms, reward structures, and expectations around care work and collegiality that often disproportionately burden women [45–51]. Moreover, the COVID-19 pandemic amplified these inequities, disproportionately impacting female academics' research time and productivity [52]. Together, these intersecting structural, relational, and cultural influences help explain the persistent gender imbalance in authorship, and highlight the need for more nuanced, intersectional, and mixed-method research to explore these dynamics in depth

If most *Veterinary Surgery* authors are ECVS or ACVS diplomates (or residents), then the disparity in female representation within ECVS and ACVS memberships may explain some of the disparity in author gender. A comparison of the ECVS/ACVS demographics with the broader veterinary profession raises particularly intriguing questions about why women are underrepresented as surgery specialists. Despite prominent feminisation of the profession, occupational segregation seems to stem from the decision to apply for a residency, rather than gender bias during the residency selection process [53], with female graduates less likely to apply for residencies to become specialists compared with their male counterparts [54]. Reluctance of women to apply for residencies may be influenced by specialty-related reputation or stereotypes in the veterinary field, which can be formative in students' professional identity [54–60]. A recent study found 47.5% of the veterinary community associated the role of surgeon with being male, while only 4.8% associated the role with being female, and 47.7% viewed it as applicable to both male and female genders [27]. The study also revealed that the most common adjectives used to describe surgeons included 'arrogant,' 'confident,' and 'egotistical'. Female graduates may also be discouraged from applying for surgical residencies because they believe surgical careers to be incompatible with family life and a balanced lifestyle [61]. Indeed, ACVS and ECVS training requirements are rigorous: programmes are to be completed within 6 years and there is no standard programme available for individuals who wish to train part-time to accommodate a family. An extension of the training period for maternity leave (note that this is not worded as 'parental' or 'adoption' leave, furthering gender bias) is mentioned in the ECVS Training Brochure (2021), but approval is required from the Credentials Committee. There is no mention of parental leave in the ACVS Residency Training Standards and Requirements (July 2024-June 2025). Both colleges may therefore be inadvertently perpetuating gendered discrimination. Finally, surgery may be off-putting for women because of well-publicised reports of sexual misconduct in human healthcare, with 90% of female surgeons within the NHS saying that they had witnessed sexual misconduct at work in the last 5 years [62]. Such detailed analyses in veterinary surgery are woefully lacking, but gender-based discrimination and sexual harassment are commonly reported by veterinary students and academic veterinarians [63,64]. Therefore, women may be reluctant to apply for surgical residencies because of an undesirable culture based around aggressive stereotypes, inflexible training, and reports of gender discrimination and sexual harassment.

Female authors were significantly underrepresented in the 'Small Animal: Orthopedics' category, which is stereotyped as a more 'masculine' discipline. This finding aligns with broader trends in STEMM fields, where women are often underrepresented in specialties associated with higher status and perceived as male-dominated [65]. The lack of women in orthopedics in ECVS membership and in publications about small animal orthopedics is stark. In human medicine, only

7% of orthopedic consultants in the UK and USA are female [66–68]. The reasons for underrepresentation of women in this field include microaggressions in the workplace, absence of flexible training options, perception of the need for physical strength, and few strong mentors and role models (both male and female) [67,68]. Similar widespread research has not yet been completed in veterinary medicine, but anecdotally the same barriers exist for female veterinary orthopedic surgeons [69]. The importance of the "trailblazer" phenomenon (whereby the rate of increase of female orthopedic surgeons is highest in areas with more preexisting female orthopaedic surgeons) [66] cannot be understated, and undergraduate and ECVS/ACVS residency programs should showcase an early positive exposure to orthopedics.

Encouragingly, the 'Large Animal' category did not show a significant author gender disparity, disproving our hypothesis that women would be underrepresented as authors in this field. This is likely a type II error because of small number of papers ('Large animal' represented under 30% of the total publications in *Veterinary Surgery* between 2002–2023) and diplomates (only 30% of ECVS/ACVS colleges specialize in Large Animal surgery) [70], but may be an outcome of genuine gender equity and vocal role modelling in this discipline [71,72].

To promote equity in authorship, journals and institutions have a vital role to play, Key strategies include efforts to combat implicit biases in the peer review process, promote diversity initiatives for editorial boards, and encourage mentorship programs [73–75]. Firstly, editorial policies that encourage double-blind review, require structured contributor role taxonomies (e.g., CRediT), and promote transparency in authorship guidelines may help mitigate implicit bias [76,77]. Inappropriate or unsubstantiated reviewer comments should be actively addressed by editors, and in some cases, reviewers may need to be removed from the process to preserve integrity and protect marginalised voices [78]. Journals must provide clear guidance to reviewers and ensure that editorial oversight mechanisms are robust enough to challenge unprofessional reviews when they occur. Without such safeguards, important conversations around inclusion and representation risk being suppressed. Secondly, editorial boards also play a critical gatekeeping role in publication outcomes. In some scientific disciplines, the gender composition of editorial teams may shape decisions on manuscript acceptance, reviewer selection, and invitation to contribute [76,77,79]. Some veterinary journals make demographic data about their editorial board transparent and freely available [80], whereas others report member names only [81]. If underrepresentation exists within editorial structures, this may reinforce authorship disparities [75]. *Nature* now publishes anonymised author demographic data alongside its journal metrics, providing a powerful example of how transparency can illuminate inequities in publication patterns [82,83]. This proactive approach allows patterns of exclusion or imbalance to be identified without researchers needing to retrospectively analyse individual journals. More veterinary and scientific journals should adopt similar practices to support accountability and foster equity in authorship, and further investigation into veterinary editorial practices and demographics is warranted [80,82,83]. Finally, institutional mentorship and support programmes for early-career female academics in surgical fields are essential to ensure that research can be developed and submitted in the first place, particularly given the compounded challenges of gendered stereotypes and the scarcity of role models in certain specialties [51,84].

Regardless of surgical emphasis, author order emerged as a significant predictor of author gender. Second and last authors were significantly less likely to be female, with a 17% and 30% reduction in probability, respectively. Our findings also indicate that the gender distribution of authors has not shifted towards parity over time. Importantly, author order does not always reflect contribution or seniority in a consistent way across institutions or disciplines. While last authorship is often interpreted as an indicator of supervisory or leadership roles, this is not a universal convention [79,85,86]. We recommend future studies consider contributor role statements (e.g., CRediT taxonomy) and ORCID-linked roles to more accurately reflect academic contribution and influence. The interaction between year and author order revealed that the probability of a given author being female decreased for last authors over time, albeit this effect was small. This stagnation, and potential worsening, of gender parity in senior authorship roles highlights a notable barrier for women in ascending to more senior authorship positions. We did not investigate the reasons for author gender disparity amongst senior roles, but our findings bear similarities to the 'leaky pipeline' phenomenon where women are often under-represented

in Science, Technology, Engineering, Mathematics, and Medicine (STEMM) despite interest and being equally or overly represented at the recruitment/undergraduate stage. This metaphor has been criticized because it implies a passive loss of talent [87,88]. It does not consider the systemic barriers and layers of biases that women must overcome to maintain or succeed in a STEMM career, including stereotypes, questioning of competency, the maternal wall, and social/professional isolation [89,90]. The impact of these barriers and biases extends beyond workplaces; they shape innovation and policy-making, as well as influencing patient care and safety [91–93].

## Limitations

With this study, we aimed to establish a foundational understanding of the role of women in *Veterinary Surgery* publications, providing a basis for future research. Our study design has several important limitations that need to be acknowledged. Firstly, we inferred author gender using a binary classification (male/female) based on probabilistic algorithms and manual validation of first names. While this method is commonly used in bibliometric research [94], it assumes a fixed link between names and gender, which may not reflect individuals' lived identities. Chosen names are deeply meaningful [95], and gender is not necessarily synonymous with sex assigned at birth. Our analysis therefore reflects gender, not sex, but this approach risks misclassification and excludes non-binary, transgender, and gender-diverse individuals—an important limitation given the study's focus on equity. Future studies should consider enabling self-identification or working with professional bodies to better represent gender diversity beyond the binary. Additionally, we acknowledge the potential for regional, cultural, and linguistic bias introduced by the use of name-based inference tools, particularly for names not common in Western datasets.

Secondly, we did not examine the intersectionality of gender with other identities such as race, ethnicity, socioeconomic status, and disability. Intersectionality can have a profound impact on an individual's experiences and opportunities within the veterinary profession, and future research should consider these multifaceted identities to provide a more comprehensive understanding of our professional landscape.

Finally, our study focused solely on *Veterinary Surgery*, the official journal of both the European and American Colleges of Veterinary Surgeons (ECVS and ACVS). This narrow focus was intentional, allowing us to align authorship data with known demographic information about College membership and thereby assess disparities relative to the most likely pool of authors. While this approach provides a valuable benchmark, it limits the generalisability of our findings. Future research should explore gender representation across a broader range of veterinary journals to determine whether similar patterns exist across the profession. However, such comparisons may present additional challenges, as author demographics are less easily matched to defined professional populations outside the ECVS/ACVS context.

## Conclusion

In conclusion, despite the increasing representation of women in the veterinary profession, gender disparities in authorship remain evident within *Veterinary Surgery*, particularly in senior authorship positions and in specific disciplines such as small animal orthopaedics. While our findings show association rather than causation, the patterns observed raise important questions about structural and cultural barriers to authorship equity. These are likely because of unequal access to mentorship, persistent stereotypes about leadership and competence, and institutional norms that undervalue collaborative or non-traditional career paths.

Promoting equity demands systemic reforms to how academic contributions are evaluated, how training is delivered, how editorial and peer-review processes are conducted, and how workplace cultures support or constrain inclusive participation. Transparency in authorship criteria, widespread adoption of contributor role taxonomies (such as CRediT), and improved demographic reporting by journals could all contribute to a more accountable publishing environment.

Future research should investigate these mechanisms in greater depth through qualitative interviews, institutional audits, and mentorship network analyses. By addressing not just individual behaviours but also the systems that shape opportunity, we can work towards a more inclusive and representative veterinary research community.

## Supporting information

**S1 Table. Overview of the selection criteria for the most relevant models.**
(DOCX)

## Acknowledgments

The Authors are indebted to the support of the Royal (Dick) School of Veterinary Studies, and to David Argyle, Lisa Boden and Cat Eastwood for their support. We are grateful to GENDER.ED, the University of Edinburgh-based hub for gender and sexualities studies, for their assistance in study design, and to ECVS (especially Wiebke Schmidt-Reyer) for sharing data and continually supporting the conversation. The findings of this paper were presented in part as a poster at ECVS Annual Scientific Meeting, Valencia, 2024.

## Author contributions

**Conceptualization:** Kelly L. Bowlt Blacklock, Jill R. D. Mackay, Poppy Bristow, Ishita Parakh, Alina Paczesna, Fiona Mackay, Kathryn Pratschke.

**Data curation:** Kelly L. Bowlt Blacklock, Jill R. D. Mackay, Ishita Parakh, Alina Paczesna, Fiona Mackay, Kathryn Pratschke.

**Formal analysis:** Kelly L. Bowlt Blacklock, Jill R. D. Mackay, Poppy Bristow, Ishita Parakh, Alina Paczesna, Fiona Mackay, Kathryn Pratschke.

**Funding acquisition:** Kelly L. Bowlt Blacklock, Fiona Mackay.

**Investigation:** Kelly L. Bowlt Blacklock, Ishita Parakh, Alina Paczesna, Kathryn Pratschke.

**Methodology:** Kelly L. Bowlt Blacklock, Ishita Parakh, Alina Paczesna, Fiona Mackay, Kathryn Pratschke.

**Project administration:** Kelly L. Bowlt Blacklock, Fiona Mackay.

**Resources:** Kelly L. Bowlt Blacklock, Ishita Parakh, Alina Paczesna.

**Software:** Kelly L. Bowlt Blacklock, Jill R. D. Mackay.

**Supervision:** Kelly L. Bowlt Blacklock.

**Validation:** Kelly L. Bowlt Blacklock, Ishita Parakh, Alina Paczesna, Fiona Mackay, Kathryn Pratschke.

**Visualization:** Kelly L. Bowlt Blacklock, Jill R. D. Mackay, Ishita Parakh, Alina Paczesna, Kathryn Pratschke.

**Writing – original draft:** Kelly L. Bowlt Blacklock, Jill R. D. Mackay, Poppy Bristow, Ishita Parakh, Alina Paczesna, Fiona Mackay, Kathryn Pratschke.

**Writing – review & editing:** Kelly L. Bowlt Blacklock, Jill R. D. Mackay, Poppy Bristow, Ishita Parakh, Alina Paczesna, Fiona Mackay, Kathryn Pratschke.

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
