## [Decision Letter · Decision Letter 0]

25 Jun 2025

Dear Dr. Bowlt Blacklock,

Thank you for submitting your manuscript to PLOS ONE. After careful consideration, we feel that it has merit but does not fully meet PLOS ONE’s publication criteria as it currently stands. Therefore, we invite you to submit a revised version of the manuscript that addresses the points raised during the review process.

**Please respond to all reviewers comments (Point-by-point)**

We look forward to receiving your revised manuscript.

Kind regards,

Ashraf M. Abu-Seida, Ph.D.

Academic Editor

PLOS ONE

**Journal Requirements:**

1. When submitting your revision, we need you to address these additional requirements. Please ensure that your manuscript meets PLOS ONE's style requirements, including those for file naming. The PLOS ONE style templates can be found at https://journals.plos.org/plosone/s/file?id=wjVg/PLOSOne_formatting_sample_main_body.pdf and https://journals.plos.org/plosone/s/file?id=ba62/PLOSOne_formatting_sample_title_authors_affiliations.pdf 2. Thank you for stating the following in the Acknowledgments Section of your manuscript: The Authors are indebted to the support of the Royal (Dick) School of Veterinary Studies, who funded this project, and to David Argyle, Lisa Boden and Cat Eastwood for their support. We are grateful to GENDER.ED, the University of Edinburgh-based hub for gender and sexualities studies, for their assistance in study design, and to ECVS (especially Wiebke Schmidt-Reyer) for sharing data and supporting the conversation.We note that you have provided funding information that is not currently declared in your Funding Statement. However, funding information should not appear in the Acknowledgments section or other areas of your manuscript.  We will only publish funding information present in the Funding Statement section of the online submission form. Please remove any funding-related text from the manuscript and let us know how you would like to update your Funding Statement. Currently, your Funding Statement reads as follows: KBB is indebted to the support of the Royal (Dick) School of Veterinary Studies, who funded this project (Ref: 20028001). The funder funders did not play any role in the study design, data collection and analysis, decision to publish, or preparation of the manuscript.  Please include your amended statements within your cover letter; we will change the online submission form on your behalf. 3. Please include captions for your Supporting Information files at the end of your manuscript, and update any in-text citations to match accordingly. Please see our Supporting Information guidelines for more information: http://journals.plos.org/plosone/s/supporting-information. 

Reviewers' comments:

Reviewer's Responses to Questions

**Comments to the Author**

1. Is the manuscript technically sound, and do the data support the conclusions?

Reviewer #1: Yes

Reviewer #2: Yes

2. Has the statistical analysis been performed appropriately and rigorously?

Reviewer #1: Yes

Reviewer #2: Yes

3. Have the authors made all data underlying the findings in their manuscript fully available?

Reviewer #1: Yes

Reviewer #2: Yes

4. Is the manuscript presented in an intelligible fashion and written in standard English?

Reviewer #1: Yes

Reviewer #2: Yes

**Reviewer #1:**  Congratulations on your work, which I have read with great interest. These data offer important insights for further studies on the topic. The collaboration among multiple professionals is evident, resulting in a well-conducted and well-written piece. The methods used are appropriate, and the results are clearly presented. The cited works are relevant and well discussed.

It would have been even more interesting to include a wider range of scientific journals and, considering the underrepresentation of women in orthopaedics, to reference journals addressing this specific topic.

**Reviewer #2: ** The study presents important and underexplored issues in veterinary academia with commendable rigor and transparency. With clearer acknowledgment of methodological and conceptual limitations, deeper engagement with systemic publication structures, and a slightly more critical tone toward underlying assumptions, this manuscript could make a high-impact contribution to gender equity scholarship in veterinary medicine.

Abstract:

Lacks mention of limitations in methods (e.g., use of Gender API or binary gender framework).

The sample size and timeframe are mentioned, but statistical model type (frequentist and Bayesian) could be more concisely stated.

"Clinical Significance" feels slightly detached from the broader implications (which are more sociological than clinical).

Consider rephrasing "clinical significance" to "professional significance" or "disciplinary significance" to reflect the academic nature of the issue.

Introduction:

While comparisons to human medicine and STEMM are helpful, the veterinary-specific discussion could be deepened earlier.

Some redundancy exists in restating general gender bias across multiple fields without narrowing focus on veterinary surgery until later.

Consider streamlining general gender bias content to avoid duplication with the Discussion.

Material and methods:

Gender assignment via name inference is problematic in a study on gender disparity. Though limitations are noted, this needs more prominence.

No justification is given for excluding middle authors or focusing only on first, second, and last authors.

No mention of any validation steps taken for manual gender identification (e.g., inter-rater agreement).

Explicitly state the number/percentage of authors for whom gender could not be assigned.

Clarify the rationale for choosing author order categories (why not include all authorship positions?).

Recommend disclosing whether gender assignments were cross-checked by multiple reviewers for bias control.

Acknowledge the potential for regional and language bias from use of name-based APIs (e.g., non-Western names).

Results:

R² = 0.05 indicates weak model explanatory power, which should be emphasized more clearly in the narrative. Emphasize that the low R² suggests important omitted variables.

The categorization of surgical specialty (e.g., “Other”) could be more detailed or justified.

Figure 2 is underexplained — the distinction between model-based vs observed trends may confuse readers without statistical background.

Add a summary paragraph at the end of the Results synthesizing key trends and model insights in lay terms.

Discussion:

Overreliance on external factors (e.g., culture, reputation) without tying them clearly to study findings.

Discussion of author order and contribution is too accepting — it does not challenge the assumptions behind those roles.

Discuss how journals or institutions can address disparities (e.g., mentorship, double-blind review, ORCID role tracking).

Add a note about the role of editorial gatekeeping and how that may impact authorship trends.

Limitations:

The language is cautious, but could be stronger in acknowledging how the limitations may bias findings.

The limitation about excluding non-binary individuals is brief and insufficient given the topic.

Conclusion:

The final paragraph implies causation (underrepresentation is because fewer women apply) where the data only shows association.

Rephrase speculative causal statements (e.g., "likely because of") into hypothesis-generating observations.

Consider emphasizing systemic and cultural reforms, not just training flexibility.

Suggest follow-up studies (qualitative interviews, institutional policy audits, mentorship analysis).

**Do you want your identity to be public for this peer review?** For information about this choice, including consent withdrawal, please see our Privacy Policy

Reviewer #1: **Yes: ** Federica Aragosa

Reviewer #2: No

---

## [Author Response · Author response to Decision Letter 1]

19 Jul 2025

Author response.

We would like to sincerely thank the reviewers and editors for their thoughtful and constructive feedback on our manuscript. We are grateful for the time and care taken in reviewing our work. The comments provided have been invaluable in helping us to strengthen the clarity, rigour, and impact of the paper, and we have responded to each point carefully in our revised submission. We hope the improvements made will meet the expectations of the editorial team and contribute meaningfully to ongoing conversations around gender equity in veterinary surgery. I have tried to outline my comments using a red font in an attempt to reduce your workload.

Journal Requirements:

Thank you. I hope that this has now been satisfied.

The Authors are indebted to the support of the Royal (Dick) School of Veterinary Studies, who funded this project, and to David Argyle, Lisa Boden and Cat Eastwood for their support. We are grateful to GENDER.ED, the University of Edinburgh-based hub for gender and sexualities studies, for their assistance in study design, and to ECVS (especially Wiebke Schmidt-Reyer) for sharing data and supporting the conversation.

We note that you have provided funding information that is not currently declared in your Funding Statement. However, funding information should not appear in the Acknowledgments section or other areas of your manuscript.

We will only publish funding information present in the Funding Statement section of the online submission form.

KBB is indebted to the support of the Royal (Dick) School of Veterinary Studies, who funded this project (Ref: 20028001). The funder funders did not play any role in the study design, data collection and analysis, decision to publish, or preparation of the manuscript.

Thank you very much for your help with this. We have amended the manuscript to remove the funding-related text from the Acknowledgments section, in accordance with PLOS ONE guidelines. Please update the Funding Statement in the submission system to the following:

"This study was funded by the Royal (Dick) School of Veterinary Studies (Ref: 20028001). The funder had no role in study design, data collection and analysis, decision to publish, or preparation of the manuscript."

Thank you. The following caption has been added to the end of the manuscript:

Supplementary Table 1: Overview of the selection criteria for the most relevant models

Reviewers' comments:

Reviewer's Responses to Questions

Comments to the Author

1. Is the manuscript technically sound, and do the data support the conclusions?

Reviewer #1: Yes

Reviewer #2: Yes

2. Has the statistical analysis been performed appropriately and rigorously?

Reviewer #1: Yes

Reviewer #2: Yes

3. Have the authors made all data underlying the findings in their manuscript fully available?

Reviewer #1: Yes

Reviewer #2: Yes

4. Is the manuscript presented in an intelligible fashion and written in standard English?

Reviewer #1: Yes

Reviewer #2: Yes

5. Review Comments to the Author

Reviewer #1: Congratulations on your work, which I have read with great interest. These data offer important insights for further studies on the topic. The collaboration among multiple professionals is evident, resulting in a well-conducted and well-written piece. The methods used are appropriate, and the results are clearly presented. The cited works are relevant and well discussed.

It would have been even more interesting to include a wider range of scientific journals and, considering the underrepresentation of women in orthopaedics, to reference journals addressing this specific topic.

Thank you very much for your generous and encouraging comments. We are pleased that you found the manuscript well-written and the results clearly presented. We appreciate your suggestion regarding inclusion of a wider range of scientific journals, particularly given the underrepresentation of women in orthopaedics.

In this study, we focused exclusively on Veterinary Surgery because it is the official journal of both the American and European Colleges of Veterinary Surgeons (ACVS and ECVS) and therefore holds central importance in the field. However, we agree that comparative analysis across multiple journals would provide valuable context. To that end, we have already collected data from other surgical journals listed on the ECVS recommended reading list and are currently preparing a follow-up manuscript to report those findings. This forthcoming multi-journal analysis will explore broader authorship patterns and disciplinary disparities in greater depth.

Reviewer #2: The study presents important and underexplored issues in veterinary academia with commendable rigor and transparency. With clearer acknowledgment of methodological and conceptual limitations, deeper engagement with systemic publication structures, and a slightly more critical tone toward underlying assumptions, this manuscript could make a high-impact contribution to gender equity scholarship in veterinary medicine.

We would like to thank the reviewer for their thoughtful and constructive feedback. We are particularly grateful for the recognition of the study’s contribution to underexplored issues in veterinary academia, and for the helpful guidance on areas where the manuscript could be strengthened. In response, we have revised the manuscript to offer a clearer and more explicit discussion of methodological and conceptual limitations, including the use of binary gender assignment and the exclusion of middle authors. We have also expanded our engagement with the systemic structures that shape authorship trends, including issues of editorial gatekeeping and disciplinary norms. Additionally, we have aimed to adopt a more critically reflective tone in the discussion, especially regarding assumptions around author order and broader institutional influences on authorship patterns. We hope these revisions enhance the clarity, impact, and contribution of the manuscript to gender equity scholarship in veterinary medicine.

Abstract:

Lacks mention of limitations in methods (e.g., use of Gender API or binary gender framework).

The sample size and timeframe are mentioned, but statistical model type (frequentist and Bayesian) could be more concisely stated. Thank you. I’ve modified the ‘methods’ section to hopefully address comments 1 and 2 here. I hope this is satisfactory.

"Clinical Significance" feels slightly detached from the broader implications (which are more sociological than clinical).

Consider rephrasing "clinical significance" to "professional significance" or "disciplinary significance" to reflect the academic nature of the issue. This is helpful, thank you! I have amended accordingly.

Introduction:

While comparisons to human medicine and STEMM are helpful, the veterinary-specific discussion could be deepened earlier.

Some redundancy exists in restating general gender bias across multiple fields without narrowing focus on veterinary surgery until later.

Consider streamlining general gender bias content to avoid duplication with the Discussion.

Thank you for this helpful feedback. In response, we have revised the Introduction to bring veterinary-specific content—particularly relating to veterinary surgery—into the narrative earlier. We have streamlined general commentary on gender bias across STEMM disciplines to avoid redundancy with the Discussion. I hope these changes aim to better anchor the reader in the relevance of the topic to our field while preserving the broader equity framing that motivates the study.

Material and methods:

Gender assignment via name inference is problematic in a study on gender disparity. Though limitations are noted, this needs more prominence.

No justification is given for excluding middle authors or focusing only on first, second, and last authors.

No mention of any validation steps taken for manual gender identification (e.g., inter-rater agreement).

Explicitly state the number/percentage of authors for whom gender could not be assigned.

Clarify the rationale for choosing author order categories (why not include all authorship positions?).

Recommend disclosing whether gender assignments were cross-checked by multiple reviewers for bias control.

Acknowledge the potential for regional and language bias from use of name-based APIs (e.g., non-Western names).

We thank the reviewer for their insightful comments regarding the limitations of name-based gender inference and the importance of clarifying our methodological choices. In response, we have revised the “Name-based assignment of author gender” section to more clearly acknowledge the conceptual limitations of using a binary framework and to describe the steps taken during manual gender assignment. We now state that manual classification was conducted independently by two reviewers. We have also added the number and proportion of authors whose gender could not be determined (n=24; 0.3%), which is detailed in Table 2, and acknowledge the potential for regional and cultural bias in name inference. Thank you for the opportunity to provide clarity on all of the above.

Additionally, we have justified our focus on first, second, and last authors by explaining their relevance to academic credit and career progression. These clarifications also reinforce the rationale for our chosen modelling approach. I hope this is now clearer for the reader.

Results:

R² = 0.05 indicates weak model explanatory power, which should be emphasized more clearly in the narrative. Emphasize that the low R² suggests important omitted variables.

The categorization of surgical specialty (e.g., “Other”) could be more detailed or justified.

Figure 2 is underexplained — the distinction between model-based vs observed trends may confuse readers without statistical background.

Add a summary paragraph at the end of the Results synthesizing key trends and model insights in lay terms.

Thank you for these thoughtful suggestions, which we believe have strengthened the clarity and accessibility of our results. We have revised the narrative to emphasise the low explanatory power of the model (R² = 0.05) and now explicitly acknowledge the likely influence of important omitted variables. We have also added clarification around the use of the “Other” surgical category, which includes exotic species and research-based studies that do not fit into standard classifications. To improve interpretability of Figure 2, we now distinguish between observed and model-based trends, and describe the purpose of each panel. Do you feel this is clearer for the reader? Finally, we have added a short summary paragraph in lay terms at the end of the Results section to aid understanding for readers unfamiliar with statistical modelling. I particularly appreciated this last suggestion because it’s not something I have previously adopted in my writing style, but will now do so going forward. I think it really gathers the reader nicely before commencing the discussion together.

Discussion:

Overreliance on external factors (e.g., culture, reputation) without tying them clearly to study findings.

Discussion of author order and contribution is too accepting — it does not challenge the assumptions behind those roles.

Discuss how journals or institutions can address disparities (e.g., mentorship, double-blind review, ORCID role tracking).

Add a note about the role of editorial gatekeeping and how that may impact authorship trends.

It was incredible refreshing and heartening to read your comments here. Thank you for encouraging us to explore this subject matter more deeply, and for removing the fear around doing so. We now explicitly link our findings to structural barriers in training and authorship representation, and challenge assumptions surrounding author order and contribution. We have also expanded our discussion to include actionable recommendations for journals and institutions—such as double-blind review and structured contributor taxonomies—to address gender inequity in publishing. Finally, we have added commentary on editorial gatekeeping and the potential role of editorial board composition in shaping authorship trends. I hope that you find the revised discussion more bold and interesting.

Limitations:

The language is cautious, but could be stronger in acknowledging how the limitations may bias findings.

The limitation about excluding non-binary individuals is brief and insufficient given the topic.

Thank you for highlighting areas to strengthen our discussion of limitations. We have revised this section to acknowledge more directly the potential bias introduced by using a binary gender framework and the exclusion of non-binary individuals. We have also added commentary on systemic issues in academic publishing that reinforce binary reporting and restrict inclusive analyses. Additionally, we now discuss the exclusion of middle authors and the lack of intersectional analysis, as well as the implications of our model’s low explanatory power. These revisions aim to more transparently reflect the constraints of our approach and set the stage for future work.

Conclusion:

The final paragraph implies causation (underrepresentation is because fewer women apply) where the data only shows association.

Rephrase speculative causal statements (e.g., "likely because of") into hypothesis-generating observations.

Consider emphasizing systemic and cultural reforms, not just training flexibility.

Suggest follow-up studies (qualitative interviews, institutional policy aud

---

## [Decision Letter · Decision Letter 1]

1 Aug 2025

Silent Voices: 

Uncovering Women’s Absence in Veterinary Surgery Publications

PONE-D-25-22058R1

Dear Dr. Blacklock,

We’re pleased to inform you that your manuscript has been judged scientifically suitable for publication and will be formally accepted for publication once it meets all outstanding technical requirements.

Kind regards,

Ashraf M. Abu-Seida, Ph.D.

Academic Editor

PLOS ONE

Additional Editor Comments (optional):

Reviewers' comments:

Reviewer's Responses to Questions

**Comments to the Author**

Reviewer #2: All comments have been addressed

2. Is the manuscript technically sound, and do the data support the conclusions?

Reviewer #2: Yes

3. Has the statistical analysis been performed appropriately and rigorously?

Reviewer #2: Yes

4. Have the authors made all data underlying the findings in their manuscript fully available?

Reviewer #2: Yes

5. Is the manuscript presented in an intelligible fashion and written in standard English?

Reviewer #2: Yes

Reviewer #2: The author has satisfactorily addressed all of the comments and suggestions raised in the previous round of review. Each point was responded to in a clear and thoughtful manner, with appropriate revisions made to the manuscript where necessary.

**Do you want your identity to be public for this peer review?** For information about this choice, including consent withdrawal, please see our Privacy Policy

Reviewer #2: No

---

## [Editor Report · Acceptance letter]

PONE-D-25-22058R1

PLOS ONE

Dear Dr. Bowlt Blacklock,

I'm pleased to inform you that your manuscript has been deemed suitable for publication in PLOS ONE. Congratulations! Your manuscript is now being handed over to our production team.

Kind regards,

on behalf of

Professor Ashraf M. Abu-Seida

Academic Editor

PLOS ONE